# Fine-Tuning Diffusion Generative Models via Rich Preference Optimization

## Abstract

We introduce Rich Preference Optimization (RPO), a novel pipeline that leverages rich feedback signals to improve the curation of preference pairs for fine-tuning text-to-image diffusion models. Traditional methods, like Diffusion-DPO, often rely solely on reward model labeling, which can be opaque, offer limited insights into the rationale behind preferences, and are prone to issues such as reward hacking or overfitting. In contrast, our approach begins with generating detailed critiques of synthesized images, from which we extract reliable and actionable image editing instructions. By implementing these instructions, we create refined images, resulting in synthetic, informative preference pairs that serve as enhanced tuning datasets. We demonstrate the effectiveness of our pipeline and the resulting datasets in fine-tuning state-of-the-art diffusion models.

## 1 Introduction

Learning from feedback and critiques is essential for enhancing the performance of a generative model by guiding the model to rectify unsatisfactory outputs, e.g. in RLHF (Ouyang et al., 2022). However, improvements can arise not only from distinguishing right from wrong, but also from receiving thoughtful and informative feedback that offers clear direction for enhancement, which we refer as *rich feedback*. For instance, in the natural language tasks, rich feedback has proved to be useful in LLM self-refinement (Saunders et al., 2022), reward model enhancement (Ye et al., 2024; Ankner et al., 2024), code debugging (Chen et al., 2023; McAleese et al., 2024), games (Paglieri et al., 2024; Hudi et al., 2025), and agents (Shinn et al., 2023; Wang et al., 2023; Renze & Guven, 2024). Rich feedback is comparatively less explored in vision tasks, such as human feedback generation model (Liang et al., 2024) and visual commonsense reasoning (Cheng et al., 2024; Chen et al., 2024; Li et al., 2024c).

To effectively leverage rich feedback in model training, it is crucial to ensure that the rich feedback is detailed, informative, and nuanced. Simply relying on numerical scores, as in classical reward models e.g. PairRM (Jiang et al., 2023; Kirstain et al., 2023b), falls short in identifying specific areas for model enhancement. Comprehensive feedback like heat maps of unaligned parts in (Liang et al., 2024) provides insights that extend beyond numerical evaluations, allowing for more targeted and substantial model improvements. In this context, exploring the use of critic models can be highly beneficial, as these models are growing rapidly and could offer deeper insights into the model's intricacies, thus contribute to a more robust understanding of improvement opportunities.

To leverage rich feedback in preference learning, we draw inspiration from the way students learn from their teachers. We introduce **R**ich **P**reference **O**ptimization **(RPO)**, a novel approach designed to enhance preference learning of images by letting the model learn from **rich preferences** that we curate with the aid of vision-language models (VLMs). These multi-modal models provide detailed critiques for downstream tasks need, offering rich feedback that mirrors the comprehensive guidance students receive in modern educational systems. Rather than merely receiving a final score on assignments or exams, students are given specific feedback that identifies logical errors, misunderstandings, or calculation mistakes. This feedback enables students to iteratively refine their initial responses, fostering deeper learning through a process of continuous improvement. Similarly, in RPO, we extract actionable editing instructions from VLMs and

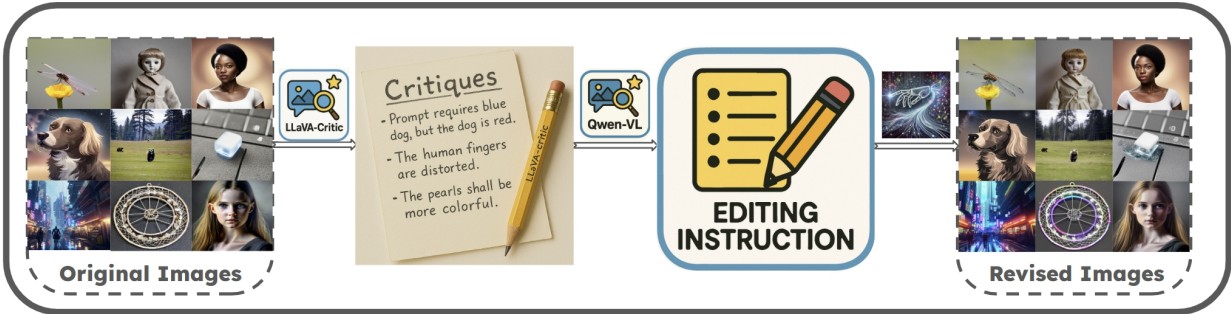

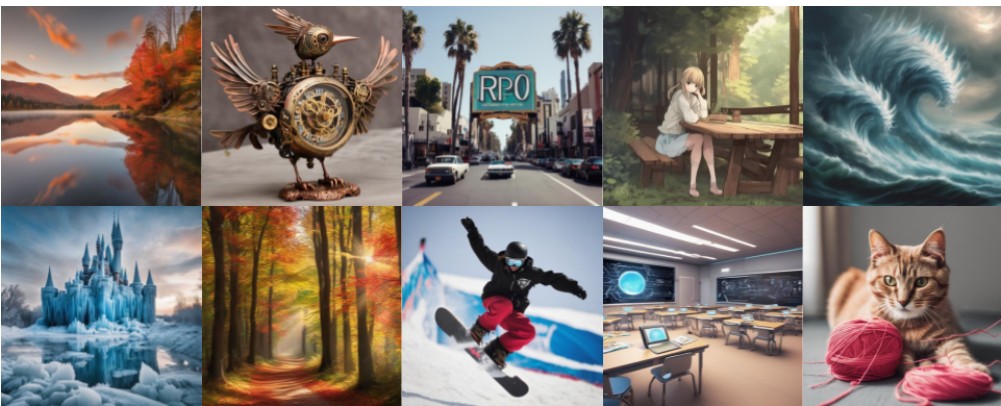

Figure 1: **(Top)** Our RPO pipeline for curating informative preference pairs from images generated from the base diffusion models: **(1)** Rich Feedback/Critic generation by a Vision Language Model (for which we choose LLaVA-Critic-7B), **(2)** Actionable editing instruction generation based on the critiques by another VLM (for which we chose Qwen2.5-VL-8B-Instruct), **(3)** Instruction-following image editing from the generated editing instructions (for which we choose ControlNet), and **(4)** Diffusion DPO training using reward model filtered synthetic preference pairs. **(Bottom)** Sample images generated from RPO fine-tuned Stable Diffusion XL, by further aligning the model on our generated synthetic preferences.

employ instruction-driven image-editing models for refinement. This scalable method generates informative preference pairs that are crucial for effective preference learning. The process of receiving detailed feedback and making refinements is akin to learning from true preference pairs, reflecting the natural and effective way in which humans learn.

The contribution of our paper is summarized as follows:

(1) We introduce RPO, a novel approach for generating preference datasets for images by leveraging VLMs to provide detailed critiques as rich feedback about misalignment between prompt and generated image. We further extract actionable editing instructions from VLMs, and employ instruction-driven image-editing models for refinement. This scalable data curation approach yields informative preference pairs, which we illustrate in Figure 1.

(2) We show that the intermediate critiques can improve the quality of editing instructions than directly generating them, which is reminiscent to the Chain-of-Thought (Wei et al., 2022) concept in mathematical reasoning for LLMs. We also propose to use ControlNet for image-editing by conditioning on the original image and combine prompts with editing instructions, which ensures fine-grained control over the image to be revised while keeping most of the image unchanged.

(3) We also empirically demonstrate that our curated preferences are scalable and useful offline synthetic preference dataset, by further training Diffusion DPO (trained on the original offline dataset) checkpoints on rich preferences we generate on the preferred images. We observe drastic performance gain, which

validates the effectiveness of our pipeline. We include qualitative samples of our obtained checkpoints also in Figure 1.

**Related Work**. In contemporary RLHF pipelines for LLMs, preference pairs are generated by sampling various responses and subsequently ranking them using either human evaluators or pretrained reward models, which serve as AI-based labels. This approach is widely utilized in both online reinforcement learning (RL)-based techniques (Ouyang et al., 2022; Bai et al., 2022) and offline methods for LLM preference learning, such as DPO (Rafailov et al., 2023), SimPO (Meng et al., 2025), and RainbowPO (Zhao et al., 2025a), which motivates Diffusion-DPO (Wallace et al., 2024) for diffusion models. See Winata et al. (2024) for a comprehensive review.

Concurrent work to ours, (Wang et al., 2025) also investigates on utilizing dedicated feedback and then passing to the editing model in the contents of an inference time scaling method. As a comparison, we focus on alignment and our pipeline has an extra procedure of generating concrete editing instructions before passing to instruction-following editing models, which directly connects two well developed literature of VLMs and Image-editing models. Our pipeline is also easier for plug-in thanks to the recent rapid progress in instruction-following image-editing capability like in GPT-4o (Hurst et al., 2024). Our work is also reminiscent to Aligner (Ji et al., 2024), which directly tries to find a more preferred answer.

## 2 Preliminaries

**Diffusion Models**. Diffusion models are a class of generative models $p_\theta(x_0)$, whose goal is to turn noisy/non-informative initial distribution $p_{\text{noise}}(x_T)$ to a desired target distribution $p_{\text{tar}}(x_0)$ through a well-designed denoising process (Ho et al., 2020; Song et al., 2021a;b). Here we adopt the discrete-time formulation of diffusion models as in Ho et al. (2020).

Given noise scheduling functions $\alpha_t$ and $\sigma_t$ (as defined in Rombach et al. (2022)), the forward process is specified by $q(x_t \mid x_s) = \mathcal{N}\left(\frac{\alpha_t}{\alpha_s} x_s, \left(\sigma_t^2 - \frac{\alpha_t^2}{\alpha_s^2}\sigma_s^2\right) I\right)$ for $s < t$. Its time-reversed process is a Markov chain parameterized by $p_\theta(x_{0:T}) = \prod_{t=1}^{T} p_\theta(x_{t-1} \mid x_t) p_{\text{noise}}(x_T)$, where

$$p_\theta(x_{t-1} \mid x_t) = \mathcal{N}\left(\mu_\theta(t, x_t), \Sigma_t I\right), \tag{1}$$

with $\mu_\theta(t, x) := \frac{\alpha_{t-1}}{\alpha_t} x_t - \left(\frac{\alpha_{t-1}}{\alpha_t} - \frac{\alpha_t \sigma_{t-1}^2}{\alpha_{t-1}\sigma_t^2}\right)\sigma_t \epsilon_\theta(t, x_t)$ and $\Sigma_t = \sigma_{t-1}^2 - \frac{\alpha_t^2 \sigma_{t-1}^4}{\alpha_{t-1}^2 \sigma_t^2}$. The model equation 1 is trained by minimizing the evidence lower bound (ELBO):

$$\min_\theta \mathcal{L}(\theta) := \mathbb{E}_{t,\epsilon,x_0,x_t}\left[\omega\left(\lambda_t\right) ||\epsilon - \epsilon_\theta(t, x_t)||_2^2\right], \tag{2}$$

where $t \sim \mathcal{U}(0, T)$, $\epsilon \sim \mathcal{N}(0, I)$, $x_0 \sim p_{\text{tar}}(x_0)$, $x_t \sim q(x_t \mid x_0) = \mathcal{N}(\alpha_t x_0, \sigma_t^2 I)$, $\lambda_t := \frac{\alpha_t^2}{\sigma_t^2}$ is the signal-to-noise ratio, and $\omega : \mathbb{R}_+ \to \mathbb{R}_+$ is some weight function. The training process equation 2 is also known as denoising score matching (see Vincent (2011) or (Tang & Zhao, 2024, Section 4.3)). It is expected that for $\theta_*$ solving the optimization problem equation 2, the model's output distribution $p_{\theta_*}(x_0) \approx p_{\text{tar}}(x_0)$, see Chen et al. (2022); Li et al. (2024a) for the theory.

**Rich Feedback**. A good critic allows the recipient to learn and improve from the feedback. In T2I generation, RichHF (Liang et al., 2024) identifies misalignment in a multimodal instruction (i.e., an image-prompt pair), and hence, enriches the feedback. A by-product is the Rich Human Feedback dataset (RichHF-18k), consisting of fine-grained scores, and misalignment image regions and text descriptions on 18K Pick-a-Pic images (Kirstain et al., 2023a). However, the Rich Feedback model and code are both not released.

As an alternative to Rich Feedback, LLaVA-Critic (Xiong et al., 2024) is an open-source VLM that is primarily developed to give evaluation of multimodal tasks. e.g., VLM-as-a-judge and preference learning. It shows a high correlation and comparable performance to proprietary GPT models (GPT-4V/4o). In our approach, we use LLaVA-Critic as an VLM-as-a-judge: the input is a text-image pair, and the output is a critic to image-prompt misalignment. Following the Chain-of-Thought concept (Wei et al., 2022), such obtained critic will be subsequently passed to an open-source LLM to provide an editing instruction. Our experiment

shows that the proposed open-source critic + editing pipeline yields more reliable improvements than directly querying a VLM, e.g., GPT4o, for editing instruction generation: see Section 3.2 for examples.

**Instruction-following Image Editing Models**. The image-editing part in our pipeline is related to the literature of instruction-following image-editing models. We focus on diffusion-based models in this paper due to both their advantage over autoregressive models, and that our base models for fine-tuning are also diffusion-based. The pioneer models include InstructPix2Pix (IP2P) (Brooks et al., 2023), which first enables editing from instructions that inform the model which action to perform. Follow-up works include Magic Brush (Zhang et al., 2023a), Emu Edit (Sheynin et al., 2024), UltraEdit (Zhao et al., 2025b) and HQEdit (Hui et al., 2024) by introducing additional datasets for further fine-tuning based on IP2P and enhancing the performance. Existing work like HIVE (Zhang et al., 2024) has also considered to align IP2P with human feedback to enhance generation capability.

As previously mentioned, we edit images based on ControlNet (Zhang et al., 2023b) to ensure a better coherence to the original image. ControlNet has been widely used for controlling image diffusion models by conditioning the model with an additional input image. Further applications include implementations for the state-of-the-art proprietary models like Stable Diffusion 3, Stable Diffusion XL (Esser et al., 2024) and FLUX (Labs, 2024), multi-image support extensions like ControlNet++ (Li et al., 2024b). We will stick to the original ControlNet in this paper (because the offline dataset is generated by the similar model scaled by Stable Diffusion v1.5), and leave the usage of more advanced ControlNet variants in future work.

**Diffusion-DPO**. Direct Preference Optimization (DPO) (Rafailov et al., 2023) is an effective approach for learning from human preference for language models. Wallace et al. (2024) proposed Diffusion-DPO, a method to align diffusion models to human preferences by directly optimizing on human comparison data $(x_0^w, x_0^l)$ given the conditional input $c$ (the prompt). Let $\beta > 0$ be the regularization parameter, and $p_{\text{ref}}(x_{0:T}|c)$ be a (pretrained) reference model. The DPO loss objective for diffusion models can be written as:

$$L_{\text{DPO}}(\theta) = -\mathbb{E}_{c,x_0^w,x_0^l} \log \sigma \left( \beta \mathbb{E}_{\substack{x_{1:T}^w \sim p_\theta(x_{1:T}^w|x_0^w,c) \\ x_{1:T}^l \sim p_\theta(x_{1:T}^l|x_0^l,c)}} \left[ \log \frac{p_\theta(x_{0:T}^w|c)}{p_{\text{ref}}(x_{0:T}^w|c)} - \log \frac{p_\theta(x_{0:T}^l|c)}{p_{\text{ref}}(x_{0:T}^l|c)} \right] \right). \tag{3}$$

By Jensen's inequality and approximate $p_\theta(x_{1:T}|x_0, c)$ by the forward process $q(x_{1:T}|x_0)$, Wallace et al. (2024) yields the following loss function for efficient training:

$$L_{\text{Diffusion-DPO}}(\theta) = -\mathbb{E}_{x_0^w,x_0^l,t,x_t^w \sim q(x_t^w|x_0^w),x_t^l \sim q(x_t^l|x_0^l)} \log \sigma \left( \beta T \omega(\lambda_t) \right. \tag{4}$$

$$\left( -\|\epsilon^w - \epsilon_\theta(x_t^w,t)\|_2^2 + \|\epsilon^w - \epsilon_{\text{ref}}(x_t^w,t)\|_2^2 + (\|\epsilon^l - \epsilon_\theta(x_t^l,t)\|_2^2 - \|\epsilon^l - \epsilon_{\text{ref}}(x_t^l,t)\|_2^2) \right),$$

where $x_t^{w,l} \sim q(x_t^{w,l} \mid x_0^{w,l}) = \mathcal{N}(\alpha_t x_0^{w,l} \sigma_t^2 I)$, and $\epsilon^{w,l} \in \mathcal{N}(0,I)$.

## 3 Curating Preference Pairs with Rich Feedback Signals

In this section, we present the concrete components in our pipeline for creating preference pairs. We utilize 1.6k rows of prompts and images from the test set of RichHF dataset provided by Liang et al. (2024) as the validation set for ablation study. We first discuss instruction-following editing (despite it being the last part before preference tuning in our pipeline), and then use the best instruction-following image-editing model we find for further ablation study on the first two components: utilizing multimodal models/VLMs for generating feedback information, and providing concise and actionable editing instructions.

### 3.1 Instruction-Following Image Editing

Despite the numerous proposed image-editing methods, either based on diffusion models or not in the literature, we find that they struggle to perform fine-grained control to follow specific editing instructions, which is crucial to generating images that are direct improvements over the existing ones. Existing methods usually change the image to another one which may yield higher score but looks fundamentally different.

To tackle the above issue, we propose a ControlNet (Zhang et al., 2023b) based image-editing approach. Concretely, we exploit the image2image (pipeline) of the ControlNet by setting the conditional image to be the same as the original one, which guarantees that the changes adhere to the original image. In addition, we concatenate the prompt that generates the image with our generated editing instructions, which we find will greatly enhance the quality of the edited image generation as illustrated in Figure 2.

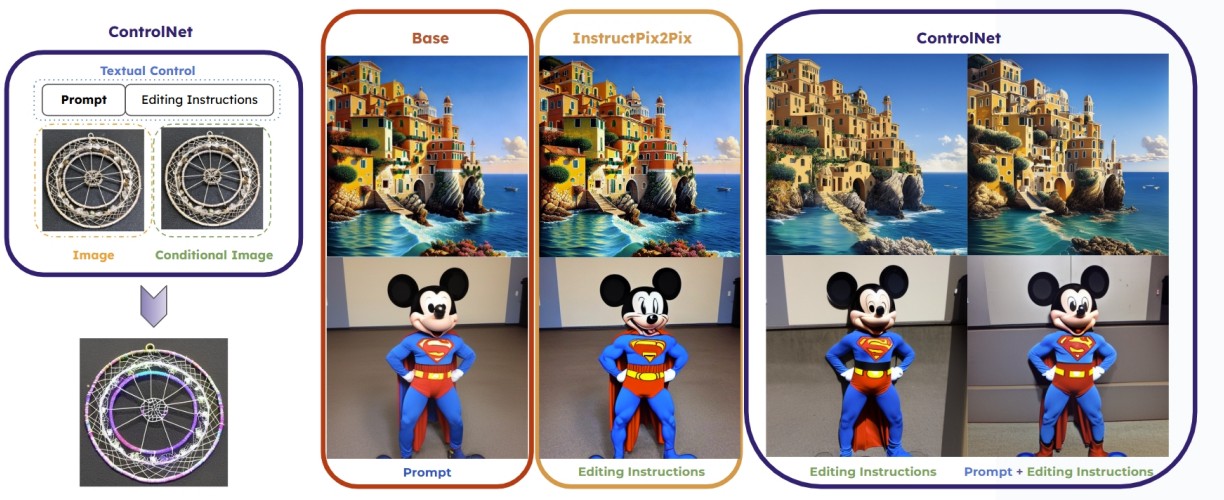

Figure 2: **(L)** We utilize ControlNet by using the same input image as the conditional image, and concatenating the prompt with the editing instruction as the textual control.
**(R)** We compare ControlNet with InstructPix2Pix and also ablate the necessity of concatenating the prompt with the editing instructions, which are generated by ChatGPT-4o. *(Top)*: The prompt is *"Italian coastline, buildings, ocean, architecture, surrealism by Michiel Schrijver."* The editing instruction is *"Incorporate iconic Italian elements like olive trees or Vespa scooters. Enhance the coastline with more distinct Mediterranean features. Add intricate architectural details typical of Italian structures. Intensify the ocean's depth with gradient blues, and ensure the surrealism reflects Michiel Schrijver's style by blending dreamlike elements."* In this case, InstructPix2Pix struggles to make any fine-grained modifications. *(Bottom)*: The prompt is *"Mickey Mouse in a Superman outfit bodybuilding."* The editing instruction is *"Adjust the character to have Mickey Mouse's face, including distinct ears, in a Superman outfit. Include bodybuilding elements such as visible muscles or weights. Ensure the outfit is accurate with the Superman logo prominently displayed."* In this case, InstructPix2Pix distorts the image. In both cases, adding the image prompt to the editing instruction for the final instruction yields better results in terms of following the instructions while keeping most of the original image unchanged.

Additionally, we quantitatively assess the instruction-following capabilities of various diffusion-based image-editing models. We also examine the impact of incorporating prompts before editing instructions, using GPT-4o for pairwise comparisons on both a standard instruction-following dataset and a test set. Our findings indicate that ControlNet-based editing is particularly effective. However, we highlight that the choice of image-editing model is flexible and can be updated to include more advanced options, such as ControlNet SDXL and other variants. We plan to explore these options in future work.

## 3.2 Generating Rich Feedback Signals

Unlike RichFB (Liang et al., 2024) which requires to train a model to detect the heatmaps and misaligned words thus providing more accurate rewards, we leverage the textual feedback on the quality of generation, like a movie critic, as in Figure 3.

To compare and ablate about what type of feedback will be more useful for yielding better editing instructions for improving the images, we prompt GPT-4o to generate them based on the given feedback, and then compute the rewards of new images (evaluated by different reward models) after using ControlNet to edit

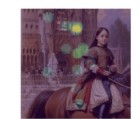

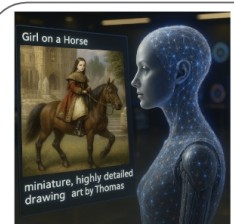

Figure 3: Comparison of RichFB generated informative feedback and our adopted textual criticism generate by carefully prompting a capable VLM.

the image with the instructions as in Figure 4a. Concretely, we let GPT-4o generate editing instructions based on various types of feedback, including those from RichFB, LLaVA-Critic, and even ChatGPT-4o itself. From the RichFB dataset, we use the image, prompt, misalignment information within the prompt, and the misalignment heatmap of the image as inputs. From LLaVA-Critic, we incorporate its textual feedback. For ChatGPT-4o, we first prompt it to also generate textual feedback based on the image-prompt pair, which is then used to derive editing instructions. For studing the necessity of rich feedback, we also let GPT 4o generate editing instructions directly from the input image and prompt, bypassing the intermediate step of critique.

Somewhat surprisingly, as in Figure 4a, we found that GPT-4o produced better editing instructions when generating them directly, rather than conditioning on its own critiques about the misalignment between the prompt and the image. However, GPT-4o generates much better editing instructions (measured by average reward) conditioning on the critiques provided by LLaVA-Critic. Based on this result, we adopt LLaVA-Critic as the feedback generator in our RPO pipeline.

Notably, here we implicitly make a hypothesis that refined images of higher reward yield better preference pairs, which also makes the corresponding VLM more preferrable. We verify this hypothesis in the experiment section 4.4 by directly comparing the trained models upon these different preference pairs, which showcases that this intuition is indeed correct.

### 3.3 Generating Editing Instructions

We also compare the performance of Llama 3.2 Vision 11B Instruct (Grattafiori et al., 2024), Llava-v1.6 Mistral 7B (Liu et al., 2024), and Qwen2.5 VL 7B Instruct (Wang et al., 2024) generated editing instructions, in combined with LLaVA-Critic and ControlNet based editing as we argued earlier. As shown in Figure 4b, Qwen2.5-VL-7B-Instruct yield the best results in leading to highest rewards in both HPSv2 and ImageReward after adopting the image-editing instructions. We thus choose Qwen2.5-VL-7B-Instruct as the VLM for editing instruction generation in our RPO pipeline.

### 3.4 Reward Model Relabeling

After obtaining the pair of original and edited images, we rearrange them into preferences by further querying a reward model or an LLM-as-a-judge. On the test set of size 16K, we find that roughly 60% of our images yield a higher score than the original image under the ImageReward (Xu et al., 2023) metric. As a remark, it is possible to use RL fine-tuning methods to encourage the VLM to generate better editing instructions,

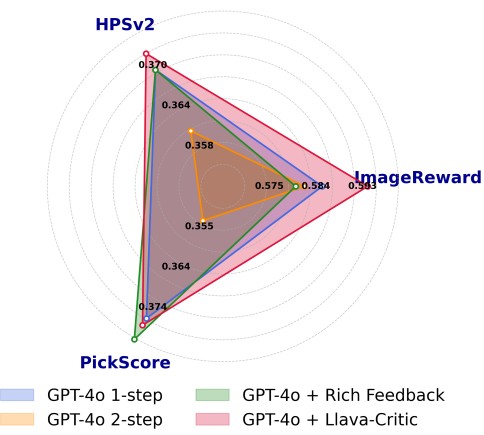

(a) Comparison of RichFB generated informative feedback and our adopted textual criticism generated by carefully prompting a capable VLM.

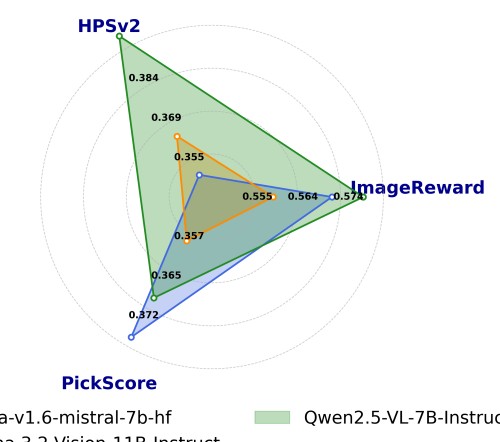

(b) Comparison of open-source VLMs combined with LLaVA-Critic in image editing quality, showing that Qwen2.5-VL-7B-Instruct is the most capable VLM model.

Figure 4: Comparisons of feedback approaches and VLM performance for enhanced image editing, evaluated by ImageReward, HPSv2 and PickScore.

which may get a higher score of edited images; we leave this for future work. Nevertheless, the edited images that fail to have higher scores than the original ones can still serve as the non-preferred images, and hence, yield the flipped preference pair.

To conclude, our RPO pipeline provides a generic and training-model-free way to generate preference pairs, as we do not need extra generations from the base model. We further use our curated dataset for fine-tuning large-scale SOTA diffusion models which validates the effectiveness of our pipeline.

## 4    Experiments

In this section, we evaluate our RPO pipeline by first fine-tuning different baseline models on our synthetic preferences data using the Diffusion-DPO algorithm, and then utilizing a couple of reward models to score the images generated by the fine-tuned models.

### 4.1    Settings

**Baseline Models.** We use SD1.5 (Rombach et al., 2022) and SDXL-1.0 (Podell et al., 2024) as our starting point. We create the following checkpoint models by Diffusion-DPO fine-tuning the two base models on the Pick-a-Pic v2 dataset: (a) DPO-SD1.5-100k, which is fine-tuned from SD1.5 using the first 100k rows; (b) DPO-SD1.5-200k, fine-tuned from SD1.5 using the first 200k rows; (c) DPO-SD1.5-100k (ImageReward-Aligned), fine-tuned from SD1.5 using the first 100k rows, with the preference modified by the relative order of ImageReward scores; (d) DPO-SDXL-100k, fine-tuned from SDXL-1.0 using the first 100k rows.

**Produced Models.** To evaluate our curated dataset, we produce the following models by Diffusion-DPO fine-tuning the baseline models using our 100k synthetic preferences data: (a) DPO-SD1.5-100k+RPO100k, which is fine-tuned from model DPO-SD1.5-100k; (b) DPO-SD1.5-100k (ImageReward-Aligned)+RPO100k, fine-tuned from model DPO-SD1.5-100k (ImageReward-Aligned); (c) DPO-SDXL-100k + RPO100k, fine-tuned from model DPO-SD1.5-100k (ImageReward-Aligned).

**Diffusion-DPO Training.** For Diffusion-DPO training, we follow the same setting and use the same hyperparameters in Wallace et al. (2024) when fine-tuning SD1.5-based models. More specifically, we use AdamW (Loshchilov & Hutter, 2019) as the optimizer; the effective batch size is set to 2048; we train at fixed square resolutions, and use a learning rate of $\frac{2000}{\beta} 2.048 \cdot 10^{-8}$ with 25% linear warmup; for the

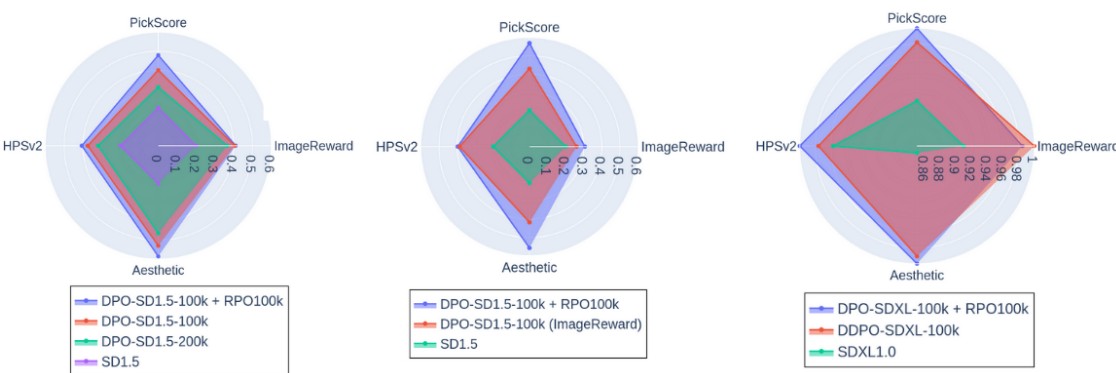

Figure 5: Model performance evaluated by PickScore, ImageReward, Aesthetic, and HPSv2.

divergence penalty parameter, we keep $\beta = 5000$. When fine-tuning SDXL-based models, we use the LoRA implementation provided by the Github project (Suzukimain & Liu, 2024) to improve training efficiency.

| Model | PickScore | ImageReward | Aesthetic | HPSv2 |
|---|---|---|---|---|
| SD1.5 | 20.33 | 0.1733 | 5.949 | 0.2622 |
| DPO-SD1.5-100k | 20.66 | 0.2784 | 6.044 | 0.2650 |
| DPO-SD1.5-200k | 20.74 | 0.3638 | 6.088 | 0.2657 |
| DPO-SD1.5-100k + RPO100k | **20.75** | **0.4395** | **6.113** | **0.2663** |
| DPO-SD1.5-100k (ImageReward-Aligned) | 20.66 | 0.600 | 6.083 | 0.268 |
| DPO-SD1.5-200k (ImageReward-Aligned) | 20.63 | 0.606 | 6.109 | 0.268 |
| DPO-SD1.5-100k (ImageReward-Aligned) + RPO100k | **20.72** | **0.686** | **6.213** | **0.269** |
| SDXL1.0 | 21.74 | 0.8473 | 6.551 | 0.2692 |
| DPO-SDXL-100k | **21.92** | 0.9183 | 6.566 | 0.2706 |
| DPO-SDXL-100k + RPO100k | 21.89 | **0.9353** | **6.585** | **0.2707** |

Table 1: Model performance on the Pick-a-Pic Test Set by four different metrics.

**Evaluation.** We generate images using the prompts from Pick-a-Pic test set (Kirstain et al., 2023b) (which contains 500 unique prompts) and evaluate the generation with reward models including PickScore (Kirstain et al., 2023b), ImageReward (Xu et al., 2023), HPSv2 (Wu et al., 2023) and LAION-Aesthetic Predictor (a ViT-L/14 CLIP model trained with SAC dataset (Pressman et al., 2022)). The PickScore, ImageReward, HPSv2 reward models are used to evaluate human-preference alignment, and Aesthetic Score is expected to evaluate visual aesthetic appeal. For each model, we report average scores over all prompts.

## 4.2 Improving Diffusion-DPO via RPO

We present the performance of all models evaluated by different reward models in Table 1. Above all, models obtained by fine-tuning on our 100k synthetic preferences data outperforms the baseline models by all metrics. This confirms our hypothesis that the pipeline we designed for synthesizing preference pairs yields improved results when combined with preference-based training algorithms such as Diffusion-DPO.

Concretely, when utilizing the *same* amount of vanilla preference data (synthetic data for RPO is curated from this same vanilla preference data), our RPO pipeline stably improves the DPO baseline over all metrics, see DPO-SD1.5-100k and DPO-SD1.5-100k+RPO100k. Even comparing with the model DPO-SD1.5-200k which utilized twice the additional vanilla preference data, our RPO fine-tuned model still performs better across all metrics. This demonstrates that our pipeline is significantly more data efficient for Diffusion-DPO style of training than just on vanilla preference data, achieving more than 100% gain over the standard practice. We have thus verified the effectiveness and potential of the synthetic preferences data generated

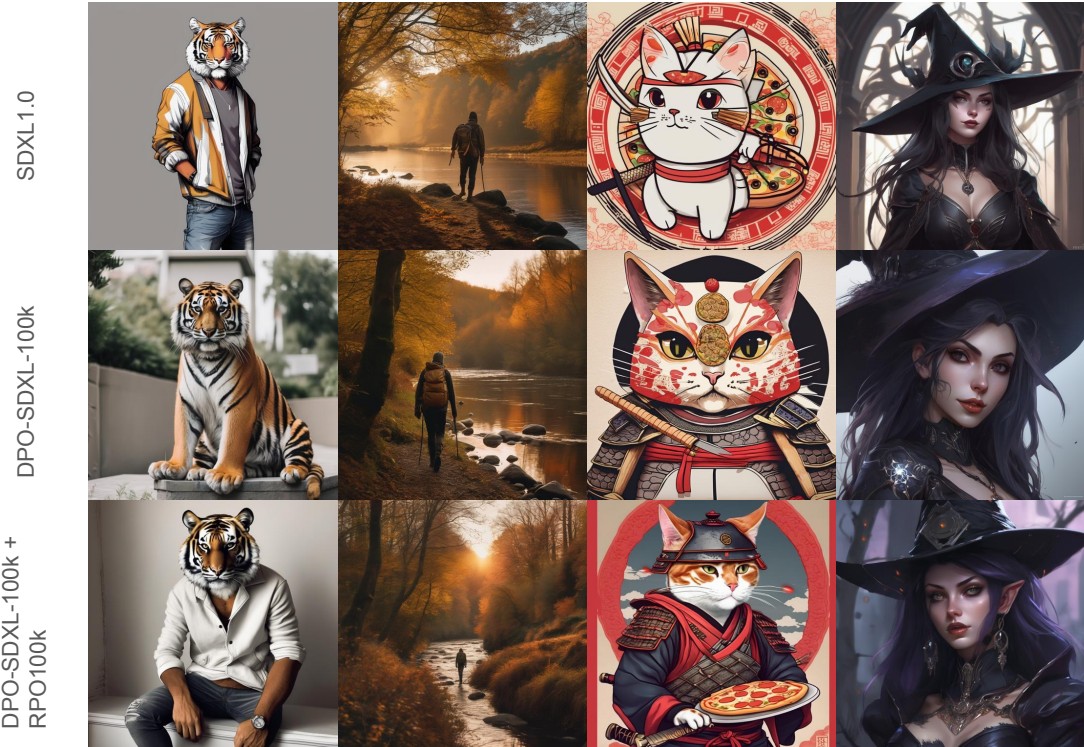

Figure 6: A comparison of generations made by SDXL1.0, DPO-SDXL-100k, and DPO-SDXL-100k+RPO100k. Prompts (from left to right): "tiger wearing casual outfit", "an adventurer walking along a riverbank in a forest during the golden hour in autumn", "samurai pizza cat", "anime portrait of a beautiful vamire witch, sci fi suit, intricate, elegant, highly detailed, digital painting, artstation, concept art, smooth, sharp focus, illustration, art by grep rutkowski and" (truncated due to the limit of number of tokens).

by rich feedback to augment the given (high-quality) dataset, which is of crucial importance when human annotated data is limited (which we believe is usually always the case in practice).

We also present a qualitative comparison of the three SDXL-based models in Figure 6. In addition to generating images with better visual qualities, we remark that the RPO fine-tuned model is better at connecting different elements in the prompts in a deep and profound way (see the first and the third columns). Understanding how rich feedback and guided revision help fine-tuning diffusion models will be left for future work.

### 4.3 Data Scaling Analysis and Generalization to Diverse Prompt Distributions

To further evaluate the robustness and scalability of our method, we conduct additional experiments on data scaling. Specifically, we subsample the vanilla preference dataset used in Diffusion-DPO training at sizes of 5K, 10K, 20K, 50K, and 100K, and compare performance between Diffusion-DPO and our Diffusion-DPO+RPO pipeline. The metrics of full Diffusion-DPO (denoted as star in Figure 7) are reported based on the evaluation of checkpoints released by Wallace et al. (2024), which is trained on nearly 1M offline preference pairs.

**Scalability.** As shown in all plots in Figure 7, DPO + RPO consistently improves upon DPO across all preference set sizes, with gains becoming increasingly pronounced as the number of training data increases. This trend holds across multiple evaluation metrics—including PickScore, ImageReward, LAION-Aesthetic, and style-specific scores (Photo, Paintings, Anime, and Concept Art), and across prompt sets such as Pick-a-Pic, PartiPrompts, and HPSv2.

**Data efficiency.** Notably, our DPO-SD1.5 model fine-tuned with only 100K vanilla preferences and enriched with 100K rich-feedback pairs (DPO-SD1.5-100k + RPO100k) outperforms the Diffusion-DPO model trained on nearly 1M Pick-a-Pic preferences on all major metrics except LAION-Aesthetic. In contrast, we observe the full DPO model exhibits inconsistent performance across reward metrics (e.g., similar ImageReward scores on Pick-a-pic and PartiPrompts and lower LAION-Aesthetic score on PartiPrompts, and more on HPSv2), suggesting overfitting to narrow preference signals. These findings highlight the strong data efficiency and generalization capabilities of RPO: it achieves superior or comparable performance using an order of magnitude less training data, without requiring additional inference or human labeling to generate new data.

**Generalization.** We further examine the generalization behavior of RPO using two out-of-distribution benchmarks: PartiPrompts and HPSv2. The PartiPrompts set evaluates transfer to novel prompt distributions, while the HPSv2 benchmark assesses robustness across diverse visual styles, including Photo, Paintings, Anime, and Concept Art. As shown in Figure 7, RPO-enhanced models consistently outperform the Diffusion-DPO baseline across all style types, prompt sets, and dataset sizes. The improvements are especially notable in the Paintings and Concept Art categories, where DPO+RPO exhibits steeper and more consistent gains as the number of preference pairs increases. The average HPSv2 score reflects this trend, with DPO+RPO maintaining a significant margin over DPO at every scale. These results highlight RPO's robustness in handling both stylistic and prompt-based distribution shifts, confirming that the benefits of rich feedback generalize effectively across domains.

### 4.4   Direct Abalations on VLM Critics

To assess the impact of different VLM critics within our RPO pipeline, we conducted a controlled ablation study in which we varied only the critic component while keeping the rest of the training pipeline fixed. Specifically, we used 1.4k prompts to generate preference pairs based on different VLM critics, including GPT-4o (1-step and 2-step), RichFeedback, and Llava-critic. We then fine-tuned models using these preference datasets and evaluated their performance across four metrics: PickScore, ImageReward, Aesthetic, and HPSv2. As summarized in Table 2, Llava-critic consistently achieved the best overall performance, with the highest scores in PickScore, ImageReward, and HPSv2. This trend is further confirmed by the radar plot in Figure 8, where Llava-critic demonstrates the most balanced and elevated performance profile. These results validate our design choice of using Llava-critic as the default VLM for generating rich feedback, given its ability to produce high-quality preference data that leads to more effective RPO fine-tuning.

| Model | PickScore | ImageReward | Aesthetic | HPSv2 |
|---|---|---|---|---|
| GPT-4o-1-step | 20.37 | 0.0684 | 6.072 | 0.2629 |
| GPT-4o-2-step | 20.41 | 0.1675 | **6.133** | 0.2635 |
| RichFeedback | 20.46 | 0.1634 | 6.076 | 0.2639 |
| Llava-critic | **20.47** | **0.2209** | 6.039 | **0.2640** |

Table 2: RPO VLM critic ablation via training on 1.4k pairs of preference data generated by different VLM critics.

### 4.5   Effectiveness of Critic-Instruction-Editing Data Curation and RPO Fine-Tuning

To evaluate the effectiveness of our curated preference data and the resulting RPO fine-tuned model, we compared images generated by the model against the ControlNet-edited images used during training, using multiple reward metrics (Table 3). The ControlNet-edited images consistently achieve higher scores, demonstrating the high quality of our training targets. Despite this, the RPO fine-tuned model yields competitive performance across all metrics, confirming the value of preference-based fine-tuning via Diffusion-DPO. More importantly, the RPO model eliminates the need for a separate critic and editing module at inference time, resulting in substantial savings in computational and memory resources. These results underscore both the quality of our critic-instruction-editing pipeline and the practical efficiency of our end-to-end generation framework.

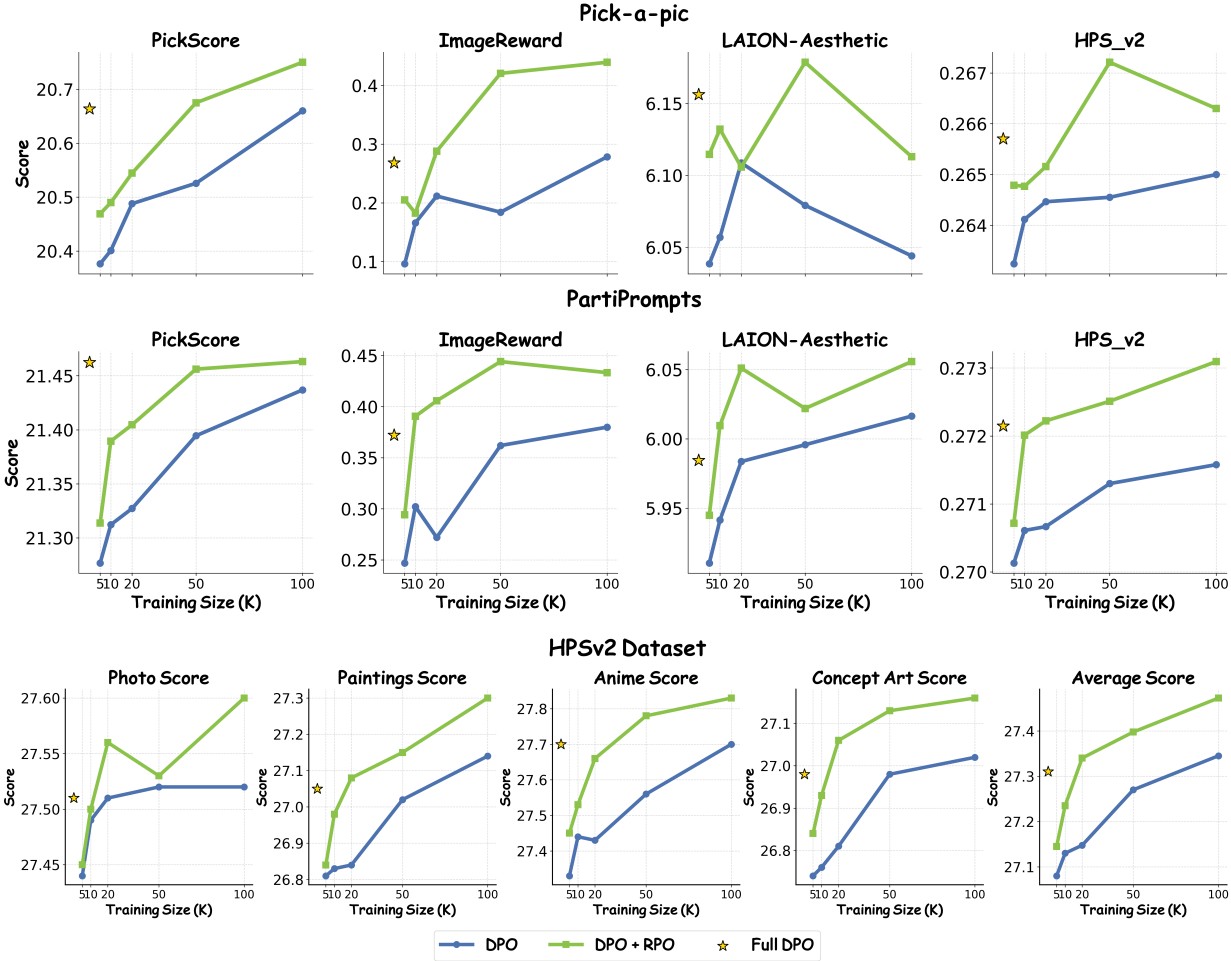

Figure 7: RPO improves DPO performance across data scales, prompt sets, and evaluation metrics. The x-axis indicates the number of Pick-a-Pic training samples used for fine-tuning. The first row presents in-distribution results on Pick-a-Pic, while the second and third rows show out-of-distribution performance on the PartiPrompts and HPSv2 prompt sets, respectively. In all plots, stars indicate the performance of the full Diffusion-DPO model, based on publicly released checkpoints from Wallace et al. (2024), which were trained on nearly 1M offline preference pairs.

| Model | PickScore | ImageReward | Aesthetic | HPSv2 |
|---|---|---|---|---|
| ControlNet | 21.61 | 0.4689 | 6.161 | 0.2813 |
| RPO | 21.02 | 0.4395 | 6.113 | 0.2663 |

Table 3: Comparison of RPO fine-tuned data and RPO fine-tuned model generation.

## 5 Discussions

Learning from AI-labeled preferences is the dominant methods in current alignment methods nowadays, but they often lack transparency since reward scores are typically black-box, and they are also not very informative, as they provide minimal insight into why a particular choice (an answer or a generated image) is preferred. Consequently, while the current preference curation pipeline is scalable, it is less efficient for aligning the model or agent based on these preferences. This inefficiency can even lead to issues like reward hacking, as noted in existing literature (Amodei et al., 2016; Denison et al., 2024).

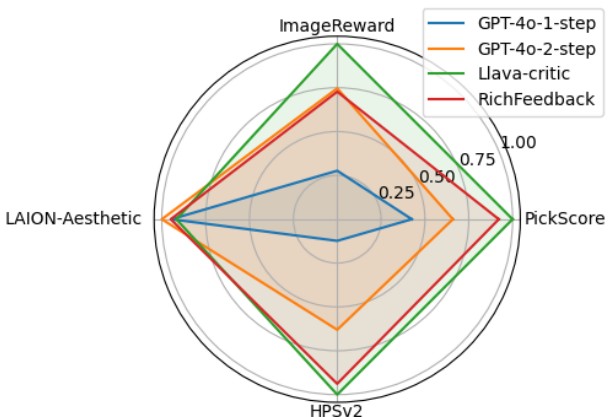

Figure 8: Radar plot comparing RPO fine-tuning performance using preference data generated by different VLM critics (GPT-4o-1-step, GPT-4o-2-step, RichFeedback, Llava-critic).

Our RPO pipeline makes a step towards addressing the challenge of creating high-quality synthetic preferences for generative models post-training. Our procedures are straighforward compliment to existing offline RLHF methods like DPO. In addition, although not explored in this paper, our pipeline can also be catered to online algorithms, such as iterative DPO or RL based methods. We consider this as a promising avenue for future research. Our pipeline also combines the existing progress and improvements in two rapidly growing research directions: VLM and image-editing models, which makes our pipeline pretty robust for future improvement by adopting more capable models as plug-ins.

## 6 Conclusion

In this work, we present **Rich Preference Optimization (RPO)**, a method that utilizes rich feedback about the prompt image alignment to improve the curation of synthetic preference pairs for fine-tuning text-to-image diffusion models. After extracting actionable editing instructions from VLMs, we employ ControlNet to modify the images, thereby producing a diverse range of refined samples. The edited images are then combined with the original versions and undergo a relabeling process using a reward model to create a curated set of preference data. By further fine-tuning checkpoint models on this synthetic dataset, we significantly enhance the performance of Diffusion-DPO training and achieve greater data efficiency. Moreover, we believe that our pipeline represents a promising direction and avenue for the data curation of synthetic vision language preference data, one that lies between two fast growing literature-VLMs and Image-editing, showcasing significant potential for future research advancements.

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

# A Input Prompts

In this section, we present the prompts used for generating critiques and editing instructions.

## A.1 Input Prompts to ChatGPT-4o for Instruction Generation

- **LLaVA-Critic feedback to editing instructions**:

> **Prompt**
>
> You are an AI assistant that provides 2-3 concise suggestions (separated by a semicolon) with each suggestion being no more than 8 words. Please make sure that each suggestion suggests concrete change, not just a high-level idea. Your goal is to improve images so they better align with the prompt: `prompt`.
>
> I want you to give short, concise editing instructions based on the following inputs regarding misalignment information. Some instructions are (1) Keep it concise: "Change the red dog to yellow" is better than "Please make the dog that is red in the image a bright yellow color". (2) Be specific: Avoid ambiguous instructions like Make it more colorful. Instead, use Change the red dog to yellow and make the background green. (3) Avoid redundancy: Don't repeat the same intent multiple times. The image is the generated image based on the prompt: `prompt`.
>
> Here, we have the feedback given by the llava critic model: fb. Please give short editing instructions for the image to solve the misalignment as a text string, where instructions are separated by a semicolon.

- **Rich feedback to editing instructions:**

> **Prompt**
>
> You are an AI assistant that provides 2-3 concise suggestions (separated by a semicolon) with each suggestion being no more than 8 words. Please make sure that each suggestion suggests concrete change, not just a high-level idea. Your goal is to improve images so they better align with the prompt: `prompt`. The first image is the generated image that we want to improve. The second image is a heatmap highlighting areas that misalign with the prompt.
>
> I want you to give short, concise editing instructions based on the following inputs regarding misalignment information. Some instructions are (1) Keep it concise: "Change the red dog to yellow" is better than "Please make the dog that is red in the image a bright yellow color". (2) Be specific: Avoid ambiguous instructions like Make it more colorful. Instead, use Change the red dog to yellow and make the background green. (3) Avoid redundancy: Don't repeat the same intent multiple times. The image is the generated image based on the prompt: `prompt`.
>
> Here, we have the list of pairs where the first element is a word in the prompt, and the second element is 1 if there's misalignment for this word, and 0 otherwise.The list of pairs are: {misalignment-pairs}. If the pairs are None, this means that this info is unavailable, please use the image and the prompt to give advice. We also have a heatmap, which is the second image attached, highlighting the misalignment area in the original (first) generated image.
> Now please give short editing instructions for the image to solve the misalignment as a text string, where instructions are separated by a semicolon.

- **ChatGPT image-prompt to editing instructions**

---

**Prompt**

You are an AI assistant that provides 2-3 concise suggestions (separated by a semicolon) with each suggestion being no more than 8 words. Please make sure that each suggestion suggests concrete change, not just a high-level idea. Your goal is to improve images so they better align with the prompt: `prompt`.

I want you to give short, concise editing instructions based on the following inputs regarding misalignment information. Some instructions are (1) Keep it concise: "Change the red dog to yellow" is better than "Please make the dog that is red in the image a bright yellow color". (2) Be specific: Avoid ambiguous instructions like Make it more colorful. Instead, use Change the red dog to yellow and make the background green. (3) Avoid redundancy: Don't repeat the same intent multiple times. The image is the generated image based on the prompt: `prompt`.

Now please give short editing instructions for the image to solve the misalignment as a text string, where instructions are separated by a semicolon.

---

- **ChatGPT image-prompt to feedback and then editing instructions**

---

**Prompt**

You are an AI assistant that helps improve a text-to-image model. Your task is to first analyze and critique whether the image aligns with the given prompt (i.e., give some feedback) and then provide 2-3 concise suggestions (separated by a semicolon) with each suggestion being no more than 8 words. Please separate the feedback and the editing instructions with an asterisk. Please make sure that each suggestion suggests concrete change, not just a high-level idea. Your goal is to improve images so they better align with the prompt: `prompt`.

I want you to first generate feedback based on the given input image (and the prompt) and then give short, concise editing instructions based on the given image and the given prompt. For the feedback, it should be a couple of sentences. Some instructions for the editing instructions are ((1) Keep it concise: "Change the red dog to yellow" is better than "Please make the dog that is red in the image a bright yellow color". (2) Be specific: Avoid ambiguous instructions like Make it more colorful. Instead, use Change the red dog to yellow and make the background green. (3) Avoid redundancy: Don't repeat the same intent multiple times. The image is the generated image based on the prompt: `prompt`.

Now please give feedback and short editing instructions for the image to solve the misalignment as a text string, where feedback and instructions are separated by an asterisk and the instructions are separated by a semicolon.

---

### A.2 Input Prompt to LLaVA-Critic for Critique Generation

> **Prompt**
>
> You are a visual-language critic. An image was generated by a diffusion model using this prompt: `[prompt]`
> Analyze the image relative to the prompt. Please:
> 1. Identify any deviations from the prompt and explain how they differ.
> 2. Note which keywords or phrases are poorly represented or misinterpreted.
> 3. Highlight any additional issues (e.g., artifacts, distortions, low detail) and their impact on quality.
>
> Provide a clear, structured critique.

### A.3 Input Prompts to VLMs for Instruction Generation

- **Generating editing instruction from Rich Feedback**

> **Prompt**
>
> You are an AI assistant providing exactly 2 to 3 concise, specific image editing suggestions (separated by semicolons), each no more than 8 words. Suggestions must describe only how to modify the *image itself* to better align with the prompt. Do not instruct changes to the text prompt.
> Formatting rules:
>
> 1. Output a single-line string with edits, separated by semicolons.
> 2. No explanations, bullet points, or extra details."
> 3. Do not repeat exact misaligned words; describe the needed visual change.
> 4. Avoid vague edits. Instead of 'Make it colorful,' say 'Turn the red dog bright yellow.'
> 5. Always generate a response unless no relevant objects exist.
> The image is generated from this prompt: `prompt`.
>
> Below is a list of (concept, flag) pairs. A flag of 0 means the image is misaligned; a flag of 1 means it is correct. For each concept flagged 0, provide one specific visual correction. List of pairs: `mis_pairs`. Output only the editing instructions in a single line. image: `base64_combined_image`.

- **Generating editing instruction from LLaVA-Critic**

**Prompt**

You are an AI assistant providing exactly 2 to 3 concise, specific image editing suggestions (separated by semicolons), each no more than 8 words. Suggestions must describe only how to modify the *image itself* to better align with the prompt. Do not instruct changes to the text prompt.
Formatting rules:

1. Output a single-line string with edits, separated by semicolons.
2. No explanations, bullet points, or extra details."
3. Do not repeat exact misaligned words; describe the needed visual change.
4. Avoid vague edits. Instead of 'Make it colorful,' say 'Turn the red dog bright yellow.'
5. Always generate a response unless no relevant objects exist.
The image is generated from this prompt: `prompt`.

Below is an image critique highlighting deviations from the prompt. Identify the specific visual misalignments and suggest precise edits to correct them.

Critique: `critique`.

Output only the editing instructions in a single line. image: `base64_combined_image`.

# B    Qualitative Examples

## B.1    Qualitative Examples from Feedback Mechanism–VLM Combinations

We present qualitative results from different combinations of feedback methods and VLMs, using a fixed image and prompt.

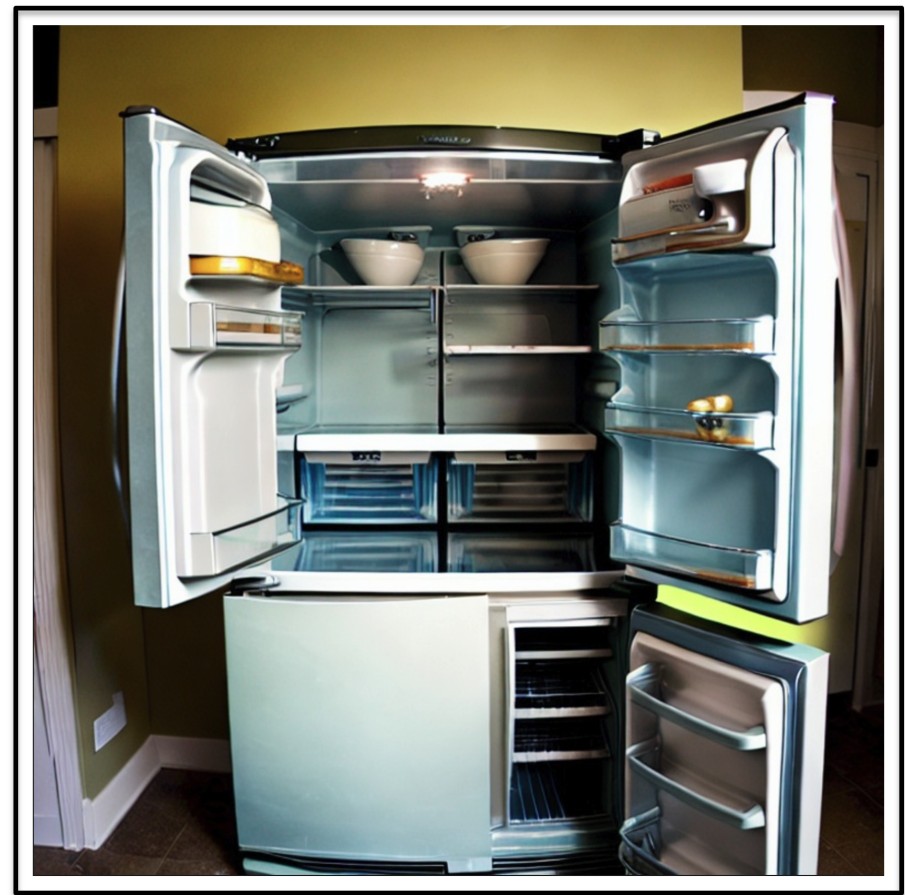

Table 9: Prompt: *An old, and broken refrigerator, open door and empty.*

**GPT-4o with Various Feedback Mechanisms.**    Table 10 shows that GPT-4o is able to generate accurate instructions to improve alignment with the prompt. However, in the 1-step generation, it enhances only two aspects in the prompt, which are *old* and *empty*. In contrast, 2-step generation, GPT-4o with Rich Feedback and GPT-4o with LLaVA-Critic successfully reinforce three elements of the prompt. In particular, GPT-4o with LLaVA-Critic's feedback produces more detailed and informative instructions.

**Rich Feedback and LLaVA-Critic with open-source VLMs.**    Tables 11 and 12 show the editing instructions generated by LLaVA, Llama and Qwen using Rich Feedback and LLaVA-Critic. We observe that LLaVA-1.6, when paired with Rich Feedback and LLaVA-Critic, can occasionally generate unhelpful instructions like "Fill empty refrigerator" and "repair the broken parts". Llama-3.2 with both feedback types, as well as Qwen-2.5 with Rich Feedback, produce reasonably good instructions but still sometimes include unnecessary parts like "lighten door" and "Add a small, empty shelf" etc. In contrast, Qwen-2.5 with LLaVA-Critic produces concise and helpful instructions that enhance three key aspects of the prompt.

| Prompt | An old, and broken refrigerator, open door and empty. | |
|---|---|---|
| GPT-4o | 1-step | Make exterior rusty and dented; Add dust on surfaces; Remove all items inside. |
| | 2-step | Add dust and dirt to surfaces; Remove all items inside; Add visible damage to fridge exterior. |
| | Rich Feedback | Add more dirt and grime; Ensure the door is open wider; Remove all items inside the fridge. |
| | LLaVA-Critic | Add rust and scratches to the exterior; cover surfaces with visible dust; make shelves appear broken and empty. |

Table 10: Qualitative examples of critiques generated by GPT-4o with different feedback generating mechanisms.

| Prompt | An old, and broken refrigerator, open door and empty. | |
|---|---|---|
| Rich Feedback | Llava-v1.6-mistral-7b-hf | Clean dusty refrigerator; Fix broken door; Fill empty refrigerator. |
| | Llama-3.2-Vison-11B-Instruct | Adjust refrigerator brightness; darken dusty areas; lighten door. |
| | Qwen2.5-VL-7B-Instruct | Add a small, empty shelf inside the refrigerator; make the refrigerator appear older and worn. |

Table 11: Qualitative examples of critiques generated by Rich Feedback with different VLMs.

| Prompt | An old, and broken refrigerator, open door and empty. | |
|---|---|---|
| LLaVA-Critic | Llava-v1.6-mistral-7b-hf | Clean the dust; repair the broken parts; remove any trash |
| | Llama-3.2-Vison-11B-Instruct | Turn the fridge light bright white; turn the floor brown; darken the edges. |
| | Qwen2.5-VL-7B-Instruct | Add dust and scratches; replace contents with old food items. |

Table 12: Qualitative examples of critiques generated by LLaVA-Critic with different VLMs.

## C   Failure Case Analysis

While our RPO pipeline produces informative preference pairs and improves downstream alignment, we also observed several systematic failure cases in data curation that limit the effectiveness of preference data pair curation. To diagnose these issues, we screened examples where the majority of reward models reported negative improvement after editing. Our analysis suggests that most failures originate in the instruction generation stage, with two recurring patterns:

- **Stray content.** Generated instructions occasionally introduce content not mentioned in the critique or even contradictory to it. For example, the instruction in Case 1 asked to replace ice cream with a giant chocolate bar, despite the original prompt containing no mention of chocolate.

- **Non-executable instructions.** In other cases, the editing instructions require targets absent in the image. For instance, in Case 2, an instruction requested "removing IKEA branding" when no such branding was present.

Beyond these instruction-level errors, we identified two additional sources of failure during the image editing stage:

- **Failure to follow instructions.** Even when the instructions are valid, the editing model sometimes diverges from them. For example, in Case 3, an instruction to correct a road sign from "TURN EFT" to "TURN LEFT" was not successfully completed by ControlNet.

- **Low editing quality in human-related images.** Edits involving human faces or bodies often introduce artifacts or unrealistic distortions, leading to degraded image fidelity. An example is shown in Case 4.

---

**Failure Case 1: Stray content**

**Prompt:**

Giant ice cream cone melting and creating a river through a city

**Critique:**

**1. Deviations from the Prompt**
The image depicts a giant ice cream cone melting and creating a river through a city, which aligns well with the prompt. However, there are a few deviations:

- **Scale**: The ice cream cone is indeed giant in scale, but it appears to be more of an artistic representation rather than a realistic one. This deviation might be intentional to create a surreal effect.

- **Melting Effect**: The melting effect is exaggerated, with the ice cream cone appearing to be melting into a river. This effect is visually striking and aligns with the prompt's description of "melting and creating a river."

**2. Keywords or Phrases Misinterpreted**

- **Giant Ice Cream Cone**: The image accurately represents this keyword, showing a massive ice cream cone.

- **Melting**: The melting effect is well-executed, though it might be slightly overdone.

- **River Through a City**: The image correctly depicts a river flowing through a city, although the city appears to be frozen or covered in snow, which adds an interesting twist.

**3. Additional Issues**

- **Artifacts and Distortions**: There are no significant artifacts or distortions visible in the image. The visual quality is high, with clear details and vibrant colors.

- **Low Detail**: The image has a high level of detail, especially in the cityscape surrounding the ice cream cone. The textures and structures of the buildings are well-defined.

**Conclusion**
Overall, the image is a creative and imaginative interpretation of the prompt. It captures the essence of the giant ice cream cone melting and creating a river through a city, with only minor deviations that enhance the surreal and artistic nature of the scene. The image is visually striking and effectively conveys the intended concept.

**Instruction:**

Replace the ice cream with a giant chocolate bar; change the river to a stream of milk.

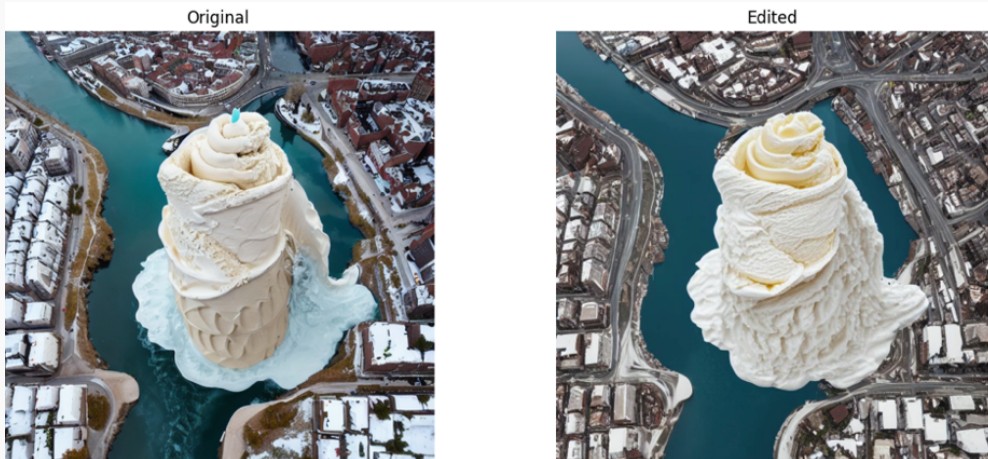

*Original (left) and Edited (right).*

**Failure Case 2: Non-executable instructions**

**Prompt:**

Detailed Ikea ad, messy Room, perfect Lighting and shadows

**Critique:**

**1. Deviations from the Prompt**

- **Messy Room**: The room appears tidy and well-organized, which contradicts the prompt's request for a "messy" room.

- **Perfect Lighting and Shadows**: The lighting in the room is indeed well-balanced and creates natural shadows, but it does not appear overly dramatic or exaggerated as one might expect in a perfect lighting scenario.

**2. Poorly Represented or Misinterpreted Keywords**

- **Ikea Ad**: The image does not clearly represent an Ikea product or advertisement. It shows a room with various items that could be associated with an Ikea style, but there are no explicit signs of being an Ikea ad.

- **Perfect Lighting and Shadows**: While the lighting is well-balanced, it does not have the dramatic effect that would make it "perfect" according to the prompt.

**3. Additional Issues**

- **Artifacts and Distortions**: There are no visible artifacts or distortions in the image.

- **Low Detail**: The image has a high level of detail, with clear textures and sharpness in the objects and furniture.

**Conclusion**
The image does not fully align with the prompt due to the room being tidy and the lighting being well-balanced rather than dramatic. The keywords "Ikea Ad" and "Perfect Lighting and Shadows" are not well-represented in the image. Overall, the image quality is high, but it does not meet the specific criteria set by the prompt.

**Instruction:**
Add cluttered furniture; enhance lighting contrast; remove IKEA branding.

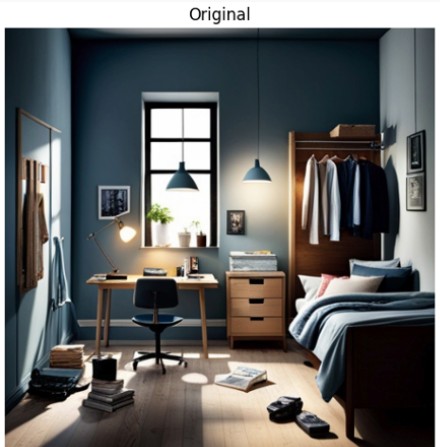 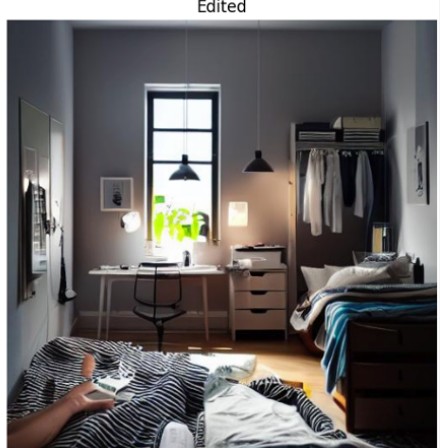

*Original (left) and Edited (right).*

**Failure Case 3: Failure to follow instructions**

**Prompt:**

A road sign with text saying "turn left for food"

**Critique:**

**1. Deviations from the Prompt**
The image provided does not match the prompt accurately. The prompt specifies a road sign with text saying "turn left for food," while the image shows a road sign that says "turn eft for food." The word "eft" is incorrect and does not align with the prompt's instruction to say "left."

**2. Keywords or Phrases Misinterpreted**

- **"Turn Left for Food"**: The prompt correctly identifies the intended message, but the image incorrectly uses "eft" instead of "left."

- **"Turn EFT for Food"**: The image uses "eft" instead of "left," which is a deviation from the prompt.

### 3. Additional Issues

- **Artifacts and Distortions**: There are no visible artifacts or distortions in the image.

- **Low Detail**: The image has a moderate level of detail, but the main issue is the incorrect wording on the sign, which affects the overall quality and accuracy.

**Conclusion**
The image deviates significantly from the prompt due to the incorrect wording on the road sign. This misinterpretation of the prompt's instructions results in a noticeable discrepancy between the expected and actual content of the image.

**Instruction:**
Correct "TURN EFT" to "TURN LEFT"; Change "FOOD" to "RESTAURANT".

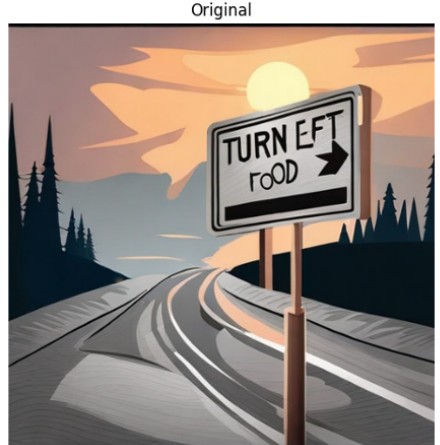 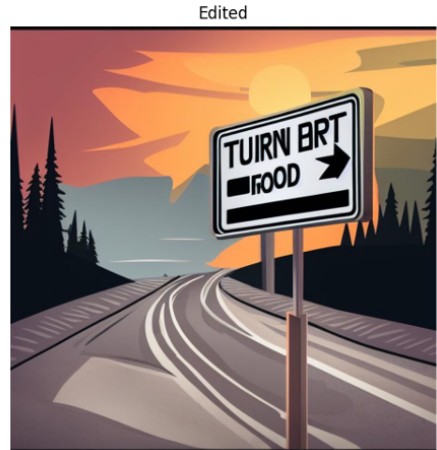

*Original (left) and Edited (right).*

---

**Failure Case 4: Low editing quality in human-related images**

**Prompt:**

photo of asian little ballet dancers resting in dance studio, nikon D5

**Critique:**

### 1. Deviations from the Prompt

- **Photography Equipment**: The image does not show any specific brand or type of camera, such as a Nikon D5, which is mentioned in the prompt.

- **Setting**: The image appears to be taken in a dance studio, but the specific details about the studio (e.g., lighting, decor) are not clearly visible or described.

- **Dancers**: The image shows four young dancers, which aligns with the prompt's mention of "little ballet dancers."

### 2. Keywords and Phrases Misinterpreted

- **Asian**: The prompt specifies "asian," but the image does not provide any information about the ethnicity of the dancers.

- **Resting**: The dancers are sitting and appear to be resting, which aligns with the prompt.

**3. Additional Issues**

- **Artifacts and Distortions**: There are no apparent artifacts or distortions in the image.

- **Detail**: The image has a clear focus on the dancers, but the background details are not very distinct.

**Conclusion**
The image generally aligns with the prompt regarding the presence of young ballet dancers in a studio setting. However, it misses specific details about the photography equipment and the exact nature of the studio environment. The prompt's mention of "asian" is not verifiable from the image provided. Overall, the image quality is good, with clear focus on the subjects, but it could benefit from more detailed context.

**Instruction:**
Add a floral backdrop behind the dancers; adjust lighting for a softer glow.

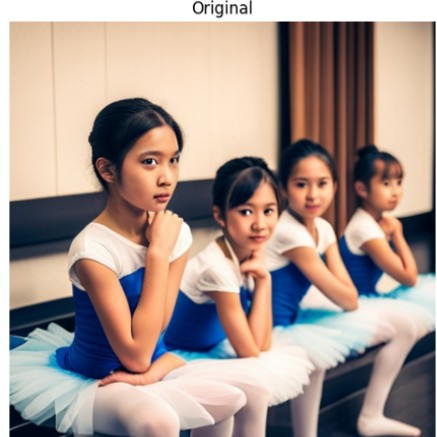 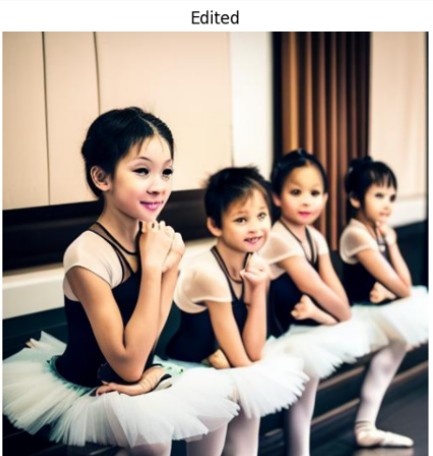

*Original (left) and Edited (right).*

