# OpenReview forum: "Fine-Tuning Diffusion Generative Models via Rich Preference Optimization"
_TMLR — Rejected by TMLR_

### Review · Reviewer_mZbv · 2025-08-25

**Summary Of Contributions:**

Contributions

The paper introduces Rich Preference Optimization (RPO), a fine-tuning framework for diffusion generative models that leverages detailed, informative and nuanced feedback rather than relying solely on scalar rewards from exiting reward models. RPO employs a pipeline involving critique generation, instruction synthesis, and image editing to create rich preference pairs, which are then used to improve model alignment. Experimental results demonstrate that RPO achieves consistent performance improvements across multiple datasets and evaluation metrics when applied on top of diffusion-DPO.

Strengths
- Clear motivation: the paper presents a well-articulated motivation for introducing Rich Preference Optimization (RPO). By emphasizing the limitations of relying solely on numerical rewards, the authors effectively make the case for using more detailed, informative and nuanced feedback during fine-tuning. The authors also drew a conceptual link between RPO and chain-of-thought reasoning, which enhances the paper's overall persuasiveness.
- Well-structured pipeline explanation: the overall RPO pipeline, including critique generation, instruction synthesis, image editing, and preference pair construction, is described in a systematic and intuitive manner. The step-by-step exposition makes it easier to follow the method.
- Competitive performance: RPO demonstrates notable performance improvements across multiple datasets and evaluation metrics, which strengthens the claim that rich feedback can lead to better-aligned diffusion models.

Weaknesses
- Evaluation relies on reward models only: while the paper motivates RPO by arguing that detailed and nuanced feedback is necessary beyond simple numerical reward scores, the evaluation for RPO fine-tuned models primarily relies on reward model based metrics such as ImageReward. This creates a mismatch between the motivation and the evaluation methodology. A more convincing analysis would include fine-grained human evaluations or region-lavel alignment assessments as in Liang et al.
- Lack of RPO-only models: All experiments apply RPO on top of diffusion-DPO fine-tuning, but there is no evaluation of RPO as a standalone fine-tuning method. Without comparisons against models fine-tuned using RPO alone, this makes RPO appear to be a marginal incremental step rather than a truly complementary method to Diffusion-DPO, as claimed.
- Scalability ablation concerns: In the scalability experiments, it is unclear whether the RPO fine-tuning uses the same dataset size as DPO or if it is fixed at 100K samples. Moreover, in Figure 7, DPO + RPO appears to outperform DPO using an "equal" amount of training data, but in fact, DPO + RPO consumes at least twice as many samples. This undermines the fairness of the reported comparison.
- Writing ambiguities in experimental setup: The paper does not clearly specify whether RPO fine-tuning always uses the same image data as DPO fine-tuning or whether additional images are introduced.
- Failure case analysis: the paper does not analyze failure cases where the newly edited images receive lower reward scores than the originals. Without detailed case studies or qualitative error analysis, it remains unclear when and why RPO pipeline fails, limiting the reliability of the reported gains.

**Audience:**

Yes

**Audience Explanation:**

Although the evaluation and analysis require further improvement, the RPO pipeline, including critique, instruction and image editing, is intuitive and compelling. I believe it is sufficient to capture the interest of the TMLR's audience.

**Broader Impact Concerns:**

The authors did not address broader impact concerns. Please add a related section in the final draft.

**Claims And Evidence:**

No

**Claims Explanation:**

Without addressing the above-mentioned weaknesses, particularly the lack of detailed and nuanced evaluation, comparisons with RPO-only models, and detailed failure case analysis, the paper does not convincingly demonstrate the superiority of RPO.

**Requested Changes:**

Please address the above weaknesses, and fix typos, e.g., compl"i"ment in Section 5.

---

> ### Author Response · Authors · 2025-09-17
>
> We sincerely thank the reviewer for providing thoughtful suggestions to improve our paper. Please find our responses to your questions below:
>
> ***Q1: Mismatch between motivation and evaluation with reward models***
>
> **A1**: We appreciate the concern and agree that optimizing a model to a single scalar reward can invite reward‑hacking. That risk is precisely why RPO uses rich, textual critiques (VLM‑as‑critic) and editing models to construct better preference pairs, not to define the objective at training time. The evaluation then reports consensus improvements across multiple, independently trained preference metrics (ImageReward, HPSv2, PickScore, Aesthetic) rather than a single score, and we measure gains on multiple prompt distributions (Pick‑a‑Pic, PartiPrompts, HPSv2) to reduce metric‑specific overfitting. Concretely, Figure 5 and Table 1 show consistent gains across all four metrics, and Figure 7 shows the same trend OOD, which is difficult to explain via hacking a single metric. Even so, we are happy to test on region-level alignment assessments as in Liang et al, if the authors can provide the reference.
>
> ***Q2: Lack of RPO-only models***
>
> **A2**: We didn’t include the results of RPO only since our primary focus in this paper is to further enhance the performance of fine-tuned diffusion models by augmenting the preference dataset instead of fully using our proposed dataset to replace the original preference data. We would pursue this comparison as future work.
>
> ***Q3: Scalability ablation concerns on fairness***
>
> **A3**: Thank you for raising this—our goal with RPO is precisely to replace expensive human supervision with cheap synthetic preference signal, not to “win” by simply seeing more human‑labeled pairs. In our current scaling plots (Fig. 7, p. 11), the x‑axis reflects the amount of human‑labeled Pick‑a‑Pic data used by Diffusion‑DPO. The RPO curve at each x‑axis value augments that same human set with synthetic, model‑generated preference pairs (critic → instruction → ControlNet edit), i.e., no additional human annotation. We will make this explicit on the figure and in the caption (and annotate total training pairs per point) to remove ambiguity.
>
> **We want to emphasize that our current comparison is fair and meaningful**. From a budget perspective, many practitioners are constrained by the number of human‑labeled pairs they can afford. Under an equal human‑label budget, RPO offers a practical path to better performance by adding synthetic preferences at near‑zero marginal human cost, due to the incremental cost of synthetic preferences in model inference (VLM critic + instruction + editor) compared to human annotation. This is already reflected in Figure 7. We also include additional trained models and their evaluation results in the updated manuscript. Even comparing with DPO-SD1.5-200k (ImageReward-Aligned) which utilized twice the precious human labelled preference dataset, our RPO model (DPO-SD1.5-100k (ImageReward-Aligned) + RPO100k) still performs better.
>
> ***Q4: Writing ambiguities in experimental setup***
>
> **A4**: We provide a more detailed description which marks red as in Section 4.2. in the updated version and included the most detailed comparisons.
>
> ***Q5: Failure Cases Analysis***
>
> **A5**: We appreciate this excellent point. We’ve conducted a screening and diagnosis on our data curation pipeline consisting of three stages: 1. Generating critiques 2. Generating instructions from the critiques 3. Editing the images. We focused on examples where the majority of reward models reported negative improvement. Our analysis indicates that most failures originate in the *instruction generation* stage, particularly due to:
>
> * Stray content (instructions that address no critique points or opposite to the critique). An example is in https://ibb.co/QjTDmK2K. The instruction asks to replace the ice cream with a giant chocolate bar, but no chocolate bar is mentioned in the original prompt.
>
> * Non-executable instructions (instructions that cannot be carried out because the original image lacks the required targets).  An example is in https://ibb.co/mC4fPR8M. The instruction requests removing the IKEA branding, but no such branding exists in the original image.
>
> Additionally, we identified two other sources of error in the editing stage:
>
> * Failure to follow the instruction: edits that diverge from the specified instruction.
>  An Example is in https://ibb.co/99VWyXn7. In this example, the instruction says changing the road sign from “TURN EFT” to “TURN LEFT”, but controlnet fails to complete the instruction.
>
> * Low editing quality in human-related images: edits involving humans often produce unnatural or unrealistic results. An Example is in https://ibb.co/zhwsZ2Tb.
>
> These findings motivate us to strengthen data curation part by employing LLMs and VLMs as validators to double-check the qualities of both instruction and editing. We greatly appreciate this suggestion!

---

### Review · Reviewer_25An · 2025-09-01

**Summary Of Contributions:**

The paper proposes Rich Preference Optimization (RPO), a pipeline for improving diffusion models by generating synthetic preference data through vision-language model critiques. Instead of relying solely on reward scores, RPO extracts detailed feedback from models, converts it into concrete editing instructions via another VLM, and uses image editors like ControlNet to revise images accordingly. These refined image pairs are then used to fine-tune diffusion models using Diffusion-DPO. Experiments show that models fine-tuned with RPO achieve better performance and data efficiency across multiple benchmarks and evaluation metrics.

**Additional Comments:**

None

**Audience:**

Yes

**Audience Explanation:**

The paper uses large VLMs to generate critiques and editing instructions, enabling scalable preference data generation for fine-tuning diffusion models. This VLM-driven approach to alignment is novel and relevant to TMLR readers interested in generative modeling, feedback learning, and multimodal systems.

**Claims And Evidence:**

No

**Claims Explanation:**

The paper provides quantitative evidence that RPO improves diffusion model generation quality using three reward models and aesthetic score. However, the assumption that “higher reward" means "better preference” isn’t always true.

**Requested Changes:**

The success of RPO relies on an implicit assumption that the base diffusion model has sufficient capacity and flexibility to learn from edited outputs. However, if the editing instructions require capabilities the model does not possess (e.g., inserting novel objects or performing semantic rearrangements), the preference supervision becomes unlearnable. The paper does not analyze these failure cases or the extent to which instruction fidelity limits learning. A critical investigation of this mismatch is necessary to assess the true generality of the approach.

---

> ### Author Response · Authors · 2025-09-17
>
> We sincerely thank the reviewer for providing thoughtful suggestions to improve our paper. Please find our responses to your questions below:
>
> ***Q1: Reward models are not equal to better preferences***
>
> **A1**: We agree that not all reward functions are equal to the real better population preferences, as actual preference is hard to obtain. This is the reason why we adopted multiple models (ImageReward, HPS v2, Aesthetic Score and PickScore) as widely used proxy to human preferences, as they are designed to capture and align with human preference. Each of these reward models has been shown to capture human preferences equally or better than a single human, as shown in Table 6 in [1]. These reward models also serve as the benchmarks in the state-of-the-art research for methodology development.
>
> [1] Wu, X., Hao, Y., Sun, K., Chen, Y., Zhu, F., Zhao, R., & Li, H. (2023). Human preference score v2: A solid benchmark for evaluating human preferences of text-to-image synthesis. arXiv preprint arXiv:2306.09341.
>
> ***Q2: Unlearnable when edits exceed model capability***
>
> **A2**: We thank the reviewer for this insightful question. We would like to justify that our pipeline is still generally employable as:
>
> **Edit magnitude is usually small in practice**. We analyzed the normalized MSE between edited and original images, and find the mean to be around 7%, indicating that edits typically modify a very small portion of the image. This matches the intent of our ControlNet setup above and supports the claim that the supervision signal lives within the model’s local support rather than demanding entirely new capabilities. As further context, §4.5 (Table 3) shows the ControlNet‑edited targets are strong according to multiple reward metrics.
>
> **Why SD‑1.5 can absorb these signals.**  Stable Diffusion 1.5 is a latent diffusion model with a VAE, UNet, and CLIP text encoder, whose expressive multi-scale architecture enables fine-grained and global edits while preserving semantics. With roughly 1.3B parameters across its components, it provides ample representational capacity, and its backbone is pre-trained on hundreds of millions of images from LAION.Such broad pretraining endows the model with sufficient flexibility to adapt to small editing signals. Fine‑tuning with RPO acts as a gentle regularizer rather than a drastic overhaul. In our pipeline, the edits themselves are produced by a ControlNet editor with SD1.5 as the backbone and is anchored to the original image (image‑to‑image with the source image as the conditioning input) and to the concatenated prompt+instruction text—see Fig. 2 (p. 5), §3.1. This keeps edits “in‑space,” i.e., local, distribution‑compatible modifications rather than open‑world hallucinations, so the resulting preference pairs are feasible learning targets.
>
> ***Q3: Failure Cases Analysis***
>
> **A3**: We appreciate this excellent point. We’ve conducted a screening and diagnosis on our data curation pipeline consisting of three stages: 1. Generating critiques 2. Generating instructions from the critiques 3. Editing the images. We focused on examples where the majority of reward models reported negative improvement. Our analysis indicates that most failures originate in the *instruction generation* stage, particularly due to:
>
> * Stray content (instructions that address no critique points or opposite to the critique). An example is in https://ibb.co/QjTDmK2K. The instruction asks to replace the ice cream with a giant chocolate bar, but no chocolate bar is mentioned in the original prompt.
>
> * Non-executable instructions (instructions that cannot be carried out because the original image lacks the required targets).  An example is in https://ibb.co/mC4fPR8M. The instruction requests removing the IKEA branding, but no such branding exists in the original image.
>
> Additionally, we identified two other sources of error in the editing stage:
>
> * Failure to follow the instruction: edits that diverge from the specified instruction.
>  An Example is in https://ibb.co/99VWyXn7. In this example, the instruction says changing the road sign from “TURN EFT” to “TURN LEFT”, but controlnet fails to complete the instruction.
>
> * Low editing quality in human-related images: edits involving humans often produce ﻿unnatural or unrealistic results. An Example is in https://ibb.co/zhwsZ2Tb.
>
> These findings motivate us to strengthen data curation part by employing LLMs and VLMs as validators to double-check the qualities of both instruction and editing. We greatly appreciate this suggestion!

---

### Review · Reviewer_2CpX · 2025-09-03

**Summary Of Contributions:**

## Conclusion of contributions
This paper proposes Rich Preference Optimization (RPO), a preference pair curation pipeline for Diffusion-DPO. It first leverages LLaVA-Critic to generate critics towards a base image and feed the critics with the base image into another VLM, such as Qwen2.5-VL, for actionable editing instruction generation. The original image is final edited based on ControlNet using the generated editing instruction and original prompt. The Diffusion-DPO trained with 100k extra RPO preference pairs shows some improvement across different prompt sets.
## Strengths
(1) The pipeline of RPO to generate better images based on the base images is well-structured and motivated. The key idea is that detailed criticisms/evaluations on images are helpful to generate better editing instructions, thus generating better images as the preferred samples.

(2) Paper is generally well written.
## Weaknesses
(1) Title is misleading: the title "Rich Preference Optimization" leads the audience to think this paper is more focused on the improvement of the preference optimization algorithm for the Diffusion model. However, the proposed method is focused on improving pairwise preference data curation.

(2) Lack of novelty: the key contribution is to introduce detailed image evaluations for better editing instructions. The other parts/modules are off-the-shelf methods. The idea that giving detailed contexts for better target generations is not fresh. Could the detailed criticisms be used in preference/RLHF optimization for the diffusion model?

(3) Experiment Weaknesses 1: In Table 1, results from many settings are missing: 1. DPO-SD1.5-RPO100k, 2. DPO-SD1.5-200k (ImageReward-Aligned), 3. DPO-SDXL-RPO100k, 4. DPO-SDXL-200k. Otherwise, only the comparison between DPO-SD1.5-200k and DPO-SD1.5-100k + RPO100k is fair.

(4) Experiment Weaknesses 2: under the current unfair baseline comparisons(smaller training set baseline VS larger training set DPO-SD+RPO), the improvement is very slight: $\textbf{1.}$ DPO-SD1.5-100k + RPO100k only significantly improved on ImageReward, less than 1% on other metrics. $\textbf{2.}$ When the baseline uses less-noised preference pairs in the ImageReward-Aligned setting, the proposed model with double training data DPO-SD1.5-100k (ImageReward-Aligned) + RPO100k is only comparable with DPO-SD1.5-100k (ImageReward-Aligned) on PickScore, Aesthetic, HPSv2. I'm curious about the result compared between DPO-SD1.5-200k (ImageReward-Aligned) and DPO-SD1.5-100k (ImageReward-Aligned) + RPO100k, also between DPO-SD1.5-100k (ImageReward-Aligned) and DPO-SD1.5-RPO100k. I believe in these two comparisons, the improvement will be even slighter. $\textbf{3.}$ when using better backbone SDXL1.0, the gap between DPO-SDXL-100k + RPO100k and DPO-SDXL-100k is also tiny, even though + RPO100k variant is trained with doubled data. Here, the results of DPO-SDXL-200k and DPO-SDXL-RPO100k are also necessary. $\textbf{4.}$ why results of ImageReward-Aligned setting of DPO-SDXL missing?

(4) Experiment Weaknesses 3: The experimental setting for Data Scaling part is not clear. Compared DPO+PRO with DPO in Figure 7, if 100k PRO data are included in DPO+PRO with different training size of original preference data, the unfair comparison results here do not make any sense.

(5) Experiment Weaknesses 4: I understand the efficiency consideration between the two manners in Table 3. But given the results in Table 3, why don't we directly use the proposed ControlNet-based pipeline to improve the quality of the zero-shot samples in a training-free manner instead of Diffusion-DPO? The performance gap in Table 3 is larger than the improvement ratio with or w/o RPO data on some metrics in Table 1.

(5) Experiment Weaknesses 5: more fine-grained metrics, such as GenEval, are necessary. Also, experiments on more advanced generators are expected.

**Audience:**

Yes

**Audience Explanation:**

The preference optimization on Diffusion models is an important topic for the community.

**Broader Impact Concerns:**

None.

**Claims And Evidence:**

No

**Claims Explanation:**

Check Experiment Weaknesses 1-4 in Summary Of Contributions.

**Requested Changes:**

Check Weaknesses in Summary Of Contributions.

---

> ### Author Response · Authors · 2025-09-17
>
> We sincerely thank the reviewer for providing thoughtful suggestions to improve our paper. Please find our responses to your questions below:
>
> ***Q1: Title is misleading: the title "Rich Preference Optimization" leads the audience to think this paper is more focused on the improvement of the preference optimization algorithm for the Diffusion model. However, the proposed method is focused on improving pairwise preference data curation.***
>
> **A1**: We agree that most existing papers on preference optimization algorithms for diffusion models have been focused on designing different loss objectives, and our current title may let readers who are familiar with aforementioned papers expect our paper to be similar. Yet our paper indeed focuses on the preference data curation part, the part where limited attention has been paid to in the literature but is of crucial importance in preference optimization. As shown in our paper, choosing good curated synthetic preference data, e.g., via the methodology we develop in the RPO pipeline, can enhance the algorithm performance as an orthogonal direction to the loss design directions. Since the model gets optimized by learning from curated rich preferences, this inspires our name of Rich Preference Optimization. We have added more clarification on this naming in the introduction part where we also mark red. We hope that our clarification will help and are also more than happy to discuss about this further.
>
> ***Q2: Lack of novelty: the key contribution is to introduce detailed image evaluations for better editing instructions. The other parts/modules are off-the-shelf methods. The idea that giving detailed contexts for better target generations is not fresh. Could the detailed criticisms be used in preference/RLHF optimization for the diffusion model?***
>
> **A2**: Our key contribution is to propose the pipeline of letting the model learn from rich preferences instead of the standard non-informative reward ranking mechanism, with rich preferences created by modifying images upon critics and editing instructions generated by VLMs. Such preference data curation mechanism and application to diffusion models fine-tuning is a fundamentally new contribution to the field to the best of our knowledge. We indeed utilized the detailed criticism to get editing instructions, which is a crucial part to get reliable preference pairs and make our RPO pipeline effective.
>
> ***Q3: Unfair baseline comparisons(smaller training set baseline VS larger training set DPO-SD+RPO)***
>
> **A3**: We would like to emphasize that our RPO utilize the same *vanilla* preference dataset as DPO baseline (i.e. DPO100K + RPO 100K vs DPO 100K). Recall that modern preference dataset like Pickapic is obtained by human labeling, making these dataset hard to reproduce or obtain, and even harder to scale up. This limitation even bottlenecked the scaling of preference tuning. RPO creates synthetic data upon the existing preference dataset, which is much cheaper to obtain by querying VLMs instead of human labellers. Also we do not even need to query additional generations of the base model which can sometimes be hard to obtain. To clarify our contribution, we are also not claiming that our synthetic dataset should replace the original preference dataset; instead, we are claiming that our synthetic dataset can augment human labelled preference data which is usually limited and scarce at almost no cost, since our pipeline is pretty scalable and generalizable. Thus, we still believe that it is quite fair in our current setup to showcase the improvement by our RPO pipeline as reported by our numbers. We provide more detailed descriptions on this in Section 4.2 and mark red.
>
> ***Q4: Additional Experiments on DPO-SD1.5-200k (ImageReward-Aligned) and DPO-SD1.5-RPO100k***
>
> **A4**: We thank the reviewers for suggestions of additional experiments. We include additional trained models and their evaluation results in the updated manuscript. Even comparing with DPO-SD1.5-200k (ImageReward-Aligned) which utilized twice the precious human labelled preference dataset, our RPO model (DPO-SD1.5-100k (ImageReward-Aligned) + RPO100k) still performs better. We didn’t include the results of RPO only since our primary focus in this paper is to further enhance the performance of fine-tuned diffusion models by augmenting the preference dataset instead of fully using our proposed dataset to replace the original preference data. We would like to pursue this direct comparison as future work.
>
> ***Q5: Directly use the proposed ControlNet-based pipeline***
>
> **A5**: We thank you for raising this suggestion. Despite it being a straightforward baseline which largely motivates our work, we do not compare with such a baseline mainly because it would require several times more inference time compute in actual deployment of the model despite being training-free, compared to our setting which requires only image generation.

---

### Decision · Action_Editor_wmPF · 2025-10-20

**Recommendation:** Reject

**Audience:**

Yes

**Audience Explanation:**

Preference optimization is an important topic in diffusion models. Researcher from this field would find the paper interesting.

**Claims And Evidence:**

No

**Claims Explanation:**

The reviewers raise several concerns that are not fully addressed in the rebuttal, including:
1. The comparison is not totally fair: The proposed method is built on top of the DPO based method, comparing only with DPO does not provide enough evidence of the advance of the proposed method. Also, the reviewers pointed out comparisons with several other settings for more fair comparisons, which are not sufficiently addressed in the rebuttal.
2. One reviewer pointed out the pipeline relies on reward models as proxies for human preference, the rebuttal addressed the concern by citing benchmarks. However, the reviewer is not satisfied because higher proxy scores does not mean better alignment in all cases.
3. Some additional experiments and analysis needs to be included in the paper to support the claim, including failure case analysis.

**Resubmission Of Major Revision:**

The authors may consider submitting a major revision at a later time.